# Patterns of Locus of Control in People Suffering from Heart Failure: An Approach by Clustering Method

**DOI:** 10.3390/medicina58111542

**Published:** 2022-10-27

**Authors:** Agnieszka Siennicka, Maciej Pondel, Szymon Urban, Ewa Anita Jankowska, Beata Ponikowska, Izabella Uchmanowicz

**Affiliations:** 1Department of Physiology and Pathophysiology, Wroclaw Medical University, 50-368 Wroclaw, Poland; 2Department of Business Intelligence in Management, Wroclaw University of Economics and Business, 53-345 Wroclaw, Poland; 3Institute of Heart Diseases, Wroclaw Medical University, 50-556 Wroclaw, Poland; 4Department of Nursing and Obstetrics, Faculty of Health Sciences, Wroclaw Medical University, 50-556 Wroclaw, Poland

**Keywords:** heart failure, patient education, clustering, noncompliance, health psychology, MHLC

## Abstract

*Background and Objectives:* The assumption of responsibility in dealing with chronic diseases is of relevance in a resource-oriented and not only deficit-oriented medicine, especially in dealing with chronic diseases, including patients with chronic heart failure. The aim of the present study is to examine, based on the model of “locus of control”, whether there are different patterns that would be relevant for a more targeted education and support of self-management in dealing with heart failure. *Materials and Methods:* For this purpose, a sample (n = 758) from 11 Polish cardiology centers have been assessed using the standardized self-assessment scale Multidimensional Health Locus of Control (MHLC), consisting of three dimensions: (i) internal localization of health control; (ii) external control by powerful others; (iii) external control by chance. *Results:* Using these three criteria, nine different clusters were extracted (mean size: 84 ± 33 patients, min 31, max 129). Three clusters included over 100 patients, whereas only two included less than 50 people. Only one cluster gathered 42 patients who will be able to cooperate with professionals in the most fruitful way. There were two clusters, including patients with beliefs related to the risk of ignoring professional recommendations. Clusters where patients declared beliefs about others’ control with low internal control should also be provided with specific help. *Conclusions:* The division into clusters revealed significant variability of belief structures about health locus of control within the analyzed group. The presented methodological approach may help adjust education and motivation to a selected constellation of beliefs as a compromise between group-oriented vs. individual approach.

## 1. Introduction

Patient education about heart failure (HF) is critical in disease management since it requires the implementation of several medical and therapeutic recommendations [1,2,3]. The effective management of HF requires not only medication adherence and frequent appointments but also regular physical activity, careful symptom monitoring (i.e., blood pressure, dyspnea, body mass, and swelling), control over the diet (including the monitoring of the intake of salt and water), and avoidance of unhealthy or risky behaviors (e.g., smoking, alcohol consumption) [1,2]. In sum, effective HF management [4,5] and doctor-patient interaction [4] depend on the patient’s knowledge about the disease [6,7,8] and their attitude towards the disease and treatment. The latest recommendations of the European Society of Cardiology (ESC) [1] for patient education in terms of self-care and symptom management are particularly demanding, and thus, their implementation is difficult and complex. The guidelines suggest that education should be patient/person-centered and encourage patient engagement in the understanding and managing their condition. Hence, it is worth looking for an optimal form of conveying education, which may help the individualization of treatment.

The education of HF patients must be aimed at delivering precise knowledge about what and how recommendations should be implemented (e.g., how to read food labels to control salt intake). Nevertheless, the education of HF patients is faced with even more challenging demands on how to motivate patients to implement all recommendations and to persist in these behaviors. Non-compliant behaviors increase the risk of worsening HF symptoms leading, to HF decompensation in severe cases [9,10,11], resulting in a serious life-threatening condition, urgent hospitalization, and faster progression of the disease in HF patients [12].

The above-mentioned elements of patient education (i.e., complex working knowledge about ways of handling the disease, implementing recommendations, self-managing health behaviors) may be supported by adequate education programs [13]. At the same time, it is well known that these elements are affected by psychological factors [14,15], including personality traits [16,17], individual beliefs, and attitudes towards the disease [18], as indicated by studies based on self-reported psychological measures [18,19,20].

The psychological characteristics of patients can predict their ability to learn and adhere to medical advice/recommendations [21]. Consequently, a psychological assessment should help tailoring recommendations more accurately on educating and motivating patients. For instance, health locus of control can be treated as one dimension in the so-called patient empowerment [22]. It has been proven that the health locus of control (defined as a set of personal beliefs about the determinants of health status) can predict health-seeking behaviors [23,24,25,26], including various forms of adherence/non-adherence [27].

In sum, for effective HF management, the contents and processes of patient education should be tailored according to a patient’s limitations, predispositions, preferences, and shared characteristics. According to the concept of multidimensional health locus of control (MHLC) [28], patients maintain regulatory beliefs about controlling their health status, which affect their behavior. For instance, patients who believe that their control matters are more likely to follow complex and demanding recommendations; whereas, patients who believe that their poor health status is just bad luck are less motivated to adhere to recommendations [29].

Interestingly, large population-based studies, based on sets of questionnaires, often ignore the individual differences of patients with chronic diseases [18,30,31] which excludes the possibility of proposing personalized recommendations for patients. The present study aims to identify structures of health-related beliefs characterizing a smaller group of HF patients (selected from a bigger sample) to personalize preparation of educational content and recommendations for further treatment using an artificial intelligence-based technique called clustering.

We decided that such a methodological approach may help select the sets of beliefs in a group of participants/respondents. Therefore, we did not intentionally form assumptions suggesting which beliefs should coexist. Thus, this study aims to apply clustering algorithms to identify patterns in the structure of beliefs related to health locus of control in patients diagnosed with HF. Recently, we have proposed a similar approach towards purely clinical data, which enables to extract phenotypes of HF patients with distinct clinical characteristics and outcomes, which should be taken into account while trial constructions and customized treatment [32]. Here, we wanted to extract behavioral phenotypes, as we believe that such a mathematical process will help to identify patients with the least and most advantageous sets of beliefs, which as a consequence, could help adjust the way of transferring information or motivating to the selected constellation of traits.

## 2. Materials and Methods

The study was based on a re-analysis of archived data, previously published in 2016 [18]. Here, we present a summary of the methodology accompanied by an explanation of the novel analytic methods applied for current analyses.

### 2.1. Study Population

The initial study group was comprised of 758 patients with systolic HF (79% males, aged 64 ± 11 years old, and a mean left ventricular ejection fraction (LVEF) of 31 ± 9%) receiving standard pharmacotherapy. The participants were recruited from a pool of hospitalized patients or patients visiting outpatient clinics in eleven cardiology centers in Poland. Patients fulfilled the following inclusion criteria: (a) >six-month documented history of HF; (b) clinical stability with unchanged medications for ≥three months preceding the study; (c) LVEF < 45%. The exclusion criteria were: (a) HF decompensation within three months preceding the study; (b) acute coronary syndrome and/or coronary revascularization within six months preceding the study; (c) any psychiatric abnormalities and/or associated therapy either at the time of examination or in the past.

The initial study, as well as the re-analysis of the data presented in the current paper, was approved by the local ethics committee.

### 2.2. Study Protocol

Data obtained using psychological questionnaires were collected during a patient’s hospital stay or his/her visit to an outpatient clinic. Additional data (e.g., age (years), sex, height, and body mass) including clinical data (e.g., resting heart rate (bpm), systolic and diastolic blood pressure (mmHg), number of years since the initial diagnosis of HF, NYHA class, HF aetiology, comorbidities, LVEF (%) determined by standard transthoracic echocardiography; basic laboratory parameters, and prescribed medications) were obtained from the medical record. Psychological evaluations were performed through administered standardized questionnaires (all Polish, officially adapted, and psychometrically validated versions) that included the MHLC Scale; Generalized Self Efficacy Scale (GSES); modified Mental Adjustment to Cancer Scale (modified Mini-MAC); Coping Inventory for Stressful Situations (CISS); and the Beck Depression Inventory (BDI). In the current paper, we report results obtained from the MHLC.

### 2.3. Instruments

The MHLC scale evaluates the psychological construct of health locus of control, which are personal beliefs on controlling an individual’s health status on three subscales [33]. The subscale of “internal localization of health control” measures how a patient’s beliefs of health status depend on its own decisions and behaviors. The subscale of “external control (by the others)” evaluates a patient’s beliefs that their health status depends on the actions of others, so-called “powerful people” (e.g., doctors, family members, and friends). The third subscale of “external control (by chance)” examines the beliefs that a patient’s health status is subject to chance, fate, or luck. Each MHLC subscale contains six items and the sum of the numerical values of the answers (6–36 scores) is based on a six-point Likert scale. The levels of agreement were worded “strongly disagree” (1 point) to the highest score “strongly agree” (6 points). There is no published guidance for establishing the overall score of health locus of control [14] and the scale can be either analyzed on an item-to-item basis (the profile analysis) or by summing all items (the aggregate analysis). MHLC subscales are independent and there is no evidence suggesting that there is a particular pattern of MHLC scores that would be the most beneficial in the context of patients’ compliance with medical regimes [29].

We decided to use MHLC, as several studies are showing that the measures of MHLC beliefs can predict compliance with medical regimes in several chronic conditions [23,34] including cardiovascular diseases [24]. Surprisingly, there is little interest in studying MHLC beliefs among HF patients, even though HF management demands several health-seeking behaviors to avoid worsening symptoms and delay the progression of the disease [1,8]. This requires, in turn, an intensive, comprehensive, and sensible communication of educational content in the field of HF management [6]. Our detailed assumptions are presented below.

### 2.4. Assumptions Related to MHLC Subscales [35]

According to the suggestions provided by the scale authors [35], utilization of the concept of health locus of control may help the individualization of treatment concerning declared beliefs [28]. In other words, variables included in the MHLC scale may serve as independent predictors of health-related behavior and treatment outcomes. For instance, studies among nurses showed that internal health control is related to a better diet, whereas a higher level of belief in control by others is linked to better self-care [36]. It was also shown that adolescents with external control of their health by chance or by other people are at increased risk for negative health outcomes [37].

There is evidence showing that the lower the level of beliefs in “chance” and the higher the beliefs in other people (e.g., doctors) the better the adherence [24]. Other studies have shown that non-adherence was significantly associated with low internal control [38]. However, there are also studies in which internal localization of health control is associated with intentional non-adherence. In contrast, external localization of health control is associated with less intentional non-adherence [27].

Importantly, the studies conducted so far show correlations between individual MHLC subscales and other variables without pointing out any interactions between beliefs within an individual patient, which may be crucial for predicting their final behavior. The MHLC subscales (especially the internal one vs. the remaining two) are independent of each other, which means that they may or may not correlate. Patients having a strong internal belief about control can, at the same time, have strong beliefs in chance and powerful others [39]. So far, there is no evidence suggesting which pattern of MHLC scores (all subscales together) are most beneficial in the context of improved health [29]. Moreover, there is practically no chance to interpret the multidimensionality of the MHLC, which reflects the multidimensionality of a human (expressing several different features at the same time). Hence, we have decided to implement the method of clustering, which will allow us to group patients according to sets of beliefs rather than a single type of belief.

### 2.5. Statistical Analysis

The assessment of normality was conducted using the Kolmogorov-Smirnov (K-S) test. Normally distributed continuous variables were presented as means ± standard deviations (SD), variables with a skewed distribution were expressed as medians with lower and upper quartiles, and categorical variables were expressed as numbers with percentages. The remaining calculations were based on the clustering approach, as described below.

### 2.6. Data Clustering

Clustering is defined as the grouping of unlabeled examples based on the mathematical similarity between them. Importantly, it is a way of natural grouping raw data and there are no prerequisites for clustering to be functional [40]. Consequently, the clustering aims to find subgroups within heterogeneous data such that each individual cluster is characterized by lower heterogeneity than the whole initial dataset [41].

To parcel out structures of MHLC beliefs into clusters, the following clustering algorithms were selected [42,43,44].
K-means based on the Euclidean distance between observations.Gaussian Mixture Model (GMM) assumes that a finite number of particular features follow a normal distribution.Agglomerative clustering using Ward’s linkage based on a classical sum-of-squares criterion produces clusters that minimize within-group dispersion at each binary fusion.

In order to select the best approach and to find the most reasonable number of clusters, we applied two well-adjusted measures of segmentation: the Davies-Bouldin index [45] and the Silhouette coefficient [46]. The Davies-Bouldin index was defined as follows:(1)$$DB=\frac{1}{n}\overset{n}{\underset{i=1}{\sum }}\frac{\sigma_{i}+\sigma_{j}}{d\left(c_{i},c_{j}\right)}\;$$
where n is the number of clusters, $\sigma_{i}$, $\sigma_{j}$ are the mean distance between elements of the cluster to the cluster centroid, and $d\left(c_{i},c_{j}\right)$ is the minimum distance of the cluster $c_{i}$ to cluster $c_{j}$ A lower Davies-Bouldin index indicates a better quality of clustering A lower Davies-Bouldin index indicates a better quality of clustering.

The Silhouette index measures how objects are similar within their cluster (cohesion) compared to other clusters (separation).
(2)$$S=\frac{1}{n}\overset{n}{\underset{i=1}{\sum }}\frac{d\left(c_{i},c\right)-d\left(c_{i}\right)}{\left\{d\left(c_{i}\right),d\left(c_{i},c\right)\right\}\;}$$
where $d\left(c_{i}\right)$ is the intra cluster distance defined as the average distance of points in cluster *i* to all other points in the cluster, $d\left(c_{i},c\right)$ is inter-cluster distance calculated as average distance of datapoints in cluster *i* to points in other clusters.

This measure evaluates the difference between the average distance within the cluster and the minimum distance between the clusters. The index ranges from −1 to +1. Segmentation results with a high Silhouette index indicate that an object has a good matching quality within the cluster and poor matching to neighboring clusters. Low and negative values indicated too few or too many clusters in the model.

The data analysis was conducted based on the following steps:
1.Selection of features for the grouping procedure.2.Visualization of the distribution in 3D space to identify potential clusters using an expert method.3.Data scaling based on the min-max method. The goal was to bring each attribute into the 0–1 domain so that, when finding clusters (calculating distances between observations), each variable had the same impact on the calculated distance.4.Conversion of 3-dimensional space into a 2-dimensional space using the PCA method and visualization of the observed distribution.5.Clustering into segments using three selected algorithms: K-means, Gaussian mixture models (GMM), and Agglomerative Clustering (AC). Each algorithm divides the dataset into 2 to 10 clusters.
(a)Assessment of cluster cohesion using the described indicators.
i.The Davies-Bouldin Index—the lowest value means the best clustering.ii.The Silhouette Coefficient—the highest value means the best clustering.(b)Selection of the optimal number of clusters and best algorithm using the Davies-Bouldin and Silhouette Coefficient indices.6.The best clustering outcomes were evaluated and interpreted using the box plots providing visualization of MHLC measures within the individual cluster and the number of patients in the clusters.

Clusters were analyzed and interpreted according to their number, the number of patients per cluster, proportions between 3 localizations of health control beliefs within all clusters, and the variability of data within each cluster.

## 3. Results

A total of 758 patients with systolic HF were recruited among 11 cardiology centers in Poland, including mainly inpatients (82%) and males (79%). The mean age of participants was 64 ± 11 years old. Sixty per cent of patients were classified with NYHA class II–III HF and had reduced ejection fractions (mean LVEF was 31 ± 9%). The aetiology of 61% of patients was ischemic HF. All patients received standard pharmacotherapy. There were the following mean scores (with SD) obtained from patients with HF: 26 ± 5 for “internal control”, 28 ± 5 for “external control (by the others)”, and 22 ± 6 for “external control (by chance)”. Detailed results reflecting the characteristics of all 11 centers separately, as well as data reflecting results from the remaining questionnaires were published in separate papers [18].

### Results of Clustering

According to the defined methodology, the prepared data has been clustered by each algorithm iteratively into various clusters, starting from 2 clusters and ending in 10 clusters. Each clustering was evaluated using both indicators to identify the optimal number of clusters and the best algorithm. The K-means algorithm achieved the highest value for the Silhouette coefficient with two clusters. The value achieved was 0.3029 (Figure 1).

To further analyze the data, the authors selected the results of K-Means clustering with 2 clusters. The distribution of MHLC-internal control, MHLC-external control, and MHLC-by chance is presented in Figure 2.

To simplify visualization, the average values of the analyzed features are presented in Figure 3. It is clear that Cluster 1 grouped patients with higher values of all three features.

When the Davies-Bouldin index was used to determine the best algorithm and the optimal number of clusters, the lowest value (the best) was achieved by the K-Means algorithm with nine clusters. The value achieved was 1.0399 (Figure 4).

The average values of the 3 analyzed variables (Y axis) in clusters 1–9 (X axis) are presented in Figure 5.

A comprehensive presentation of the 3 characteristics distributed within the nine clusters is presented in Figure 6.

The division into nine clusters revealed significant variability of structures of beliefs about health locus of control within the analyzed group. The basic parameters and comparisons with some suggestions related to future recommendations are presented in Table 1.

The mean size of the clusters was 84 ± 33 patients (min 31, max 129). Three clusters included over 100 patients, whereas only 2 included less than 50 people. Regarding the proportion of scores in each subscale reflecting 3 possible localizations of health control, cluster 1 is similar to 3 (the beliefs in other people are at the highest level with the lowest level of internal beliefs). Clusters 4, 6, and 7 were similar to 9 (where the beliefs in other people are at the highest level with the lowest level of beliefs in chance), and 5 is similar to 8 (with the highest internal beliefs and the weakest beliefs in chance). Cluster 2 is different from the remaining 8, as the dominance of internal beliefs is accompanied by an equal level of both types of external beliefs. It is important to underline that there is one cluster with an exceptionally low level of internal beliefs (cluster 3) and two clusters with an exceptionally low level of beliefs in chance (5 and 8).

This cluster included patients who believe in health control by others and who acknowledge their role in the control over their health. On the other hand, there is one problematic (2) and one very problematic (3) cluster which included 76 and 31 patients, respectively. Beliefs characterizing patients from cluster 2 may be related to the risk of ignoring professional recommendations (as beliefs in chance are equal to beliefs in others with a dominance in internal control beliefs). Cluster 3 included patients with the lowest level of internal control, which may suggest that they seriously underestimate their impact on their health. It can be presumed that such beliefs might make it harder to follow advice, especially those requiring effort and individual engagement, such as precise control over water or salt intake or regular self-monitoring. Clusters in which beliefs in others were significantly higher than beliefs in internal control may also require specific attention. Such patients may strongly trust all the recommendations provided by healthcare professionals. As a result, the patients may not only follow the recommendations but also might be disappointed with the results of their actions.

## 4. Discussion

The management of HF requires relatively extensive knowledge [1,2], and the acquisition of which is always linked to a substantial effort by the patient. HF management requires persistence in the implementation of many demanding recommendations [3]. In a perfect world, healthcare professionals should have enough time to explain to each HF patient what and how to do after leaving hospital to avoid both disease progression and re-hospitalizations. In a perfect world, healthcare professionals might even have the skills to effectively motivate the patient in a way that suits him/her (probably different approaches would be appropriate for different patients). Unfortunately, the reality is different. Usually, there is not enough time to educate patients, and no time to diversify educational content and/or form, following the requirements and preferences of the individual educated patient.

Both the education as well as ways of motivating the patient should be adapted to his or her individual abilities, limitations, and predispositions. Psychological studies conducted among patients suffering from a chronic disease (like HF) could serve not only to identify psychological phenomena accompanying somatic diseases but deliver precise guidelines on how to improve (and personalize) the help and support provided to patients. Patients with HF constitute an extremely heterogeneous group. From a practical point of view, it is worth considering looking for a phenotype not only in terms of their clinical characteristics [32], which is now an important trend in the approach to HF treatment. Perhaps (due to the complexity of recommendations addressed to the patient) it is worth considering looking for behavioral phenotypes to differentiate and/or to optimize the educational approach to these patients.

At the same time, due to the above-mentioned limitations, all educational and motivating programs are not individual, but are often addressed to a large group of patients, which excludes an individual approach. In other words, there is a need for balance between the group-oriented vs individual approach. Searching for groups of patients sharing similar mental structures may be the solution. That was the idea behind the presented analysis. We aimed to show how psychological data could be used to create specific psychological profiles of patients based on differences and similarities in features that—according to the available literature—may affect their approach to their disease, medical staff, caregivers, and overall attitude towards health-related behaviors.

This study used a clustering method to analyze and interpret the beliefs related to health locus of control within a large group of patients with HF. We selected clustering, as this method is able to select groups without any preliminary assumptions. As a result, the selection of groups was entirely based on the character of collected data, which is particularly beneficial when it comes to the evaluation of subtle mental phenomena.

This segmentation approach is novel, as existing divisions in the literature were only feasible on a binary analysis. For instance, patients were divided into subgroups with high, medium, or low severity in the psychopathological dimension (e.g., BDI revealed groups with mild, moderate, or severe depression) [47]. Similarity grouping, such as feature-based selection, is always challenging. In summary, the proposed clustering analysis indicated distinct clinically homogenous groups with variable beliefs towards health control. One could expect to observe non-comparable behavioral responses in each group to the same educational intervention. Thus, when it comes to designing educational or psychological interventions for HF patients, such findings must be considered [48,49,50] for better compliance with advice provided by healthcare professionals, self-care, and HF management. For instance, regarding probable recommendations in the view of MHLC theory, we propose that there is only one cluster (4) of 42 patients who will be able to cooperate with professionals in the most fruitful way.

The conclusion from this clustering approach for the analysis of MHLC beliefs among HF patients may be summarized as follows. The present study showed that beliefs about health locus of control in HF patients may have distinctive profile clusters. The distinguished clusters of MHLC beliefs have variable sizes. As mentioned, no data is showing which pattern of MHLC scores (using all 3 subscales together) would be most beneficial in the context of health behavior [29]. However, following the theoretical MHLC framework [29], we propose the following interpretation of our results. The applied algorithms revealed that there are 2 or 9 clusters. The division into 2 clusters showed only differences in scoring, without substantial differences in relative proportions between measured beliefs. The division into 9 clusters revealed a more significant variability of beliefs related to health control. We found that only one cluster (4) of 42 patients whose psychological characteristics (in terms of beliefs related to health control) are the most beneficial/promising in the context of compliance with advice provided by healthcare professionals. This cluster had strong beliefs in health control by others and slightly weaker beliefs in their role in the control of their health. Importantly, we detected two clusters that included patients who may require a specific approach. In cluster 2, 76 patients declared beliefs related to health control which may be linked to a risk of ignoring professional recommendations or treating them equally to non-professional sources (beliefs in chance are equal to beliefs in others with dominance in beliefs in internal control). In cluster 3, 31 patients shared the lowest level of internal control compared to all remaining clusters. This may indicate a strong tendency of patients to underestimate their impact on their health and may lead to ignoring their doctor’s advice (especially while completing tasks requiring effort and scrupulousness). Clusters with low levels of beliefs in chance (with high levels of beliefs in others) included people who theoretically are very promising in the context of compliance with advice provided by healthcare professionals as they will acknowledge the doctor’s advice without searching for justifications indicating the influence of force majeure, fate, or bad luck. However, all clusters in which beliefs in others are significantly higher than beliefs in internal control should be treated with caution. Such beliefs may be related to exceptionally high levels of trust and expectation toward healthcare professionals. Clusters of patients sharing the highest internal control (especially when internal control is stronger than the beliefs in control by others) are at risk of following individual decisions, which may not be compliant with medical guidelines [18].

Analysis of one co-occurring feature will never convey the multidimensionality of human behavior. However, the MHLC scale is multidimensional and reflects the complexity of human nature. Individual differences do not disappear when a healthy individual becomes a patient.

What are the benefits of our approach? The disease experience may affect health-related beliefs, but there is no reason to suspect that everyone will change in the same direction (which would decrease inter-individual variability). Knowing that at least some of our mental traits are stable throughout life (especially adulthood), it is worth considering whether the factor responsible for being a good or bad university student could be analogously responsible for being more or less prone to education when becoming a cardiac patient. Procrastination related to completing boring tasks or forgetting about regular car check-ups may affect behaviors, including regular blood pressure examinations or doctor’s appointments. Therefore, analyzing correlations between single psychological features may give only a partial explanation of a patient’s behavior. The analysis of a set of features is of greater value, however, it is possible only when analyzing a single patient, not a group. Clustering may serve as an attractive compromise.

In summary, we have highlighted the diversity in regard to the constellation of beliefs about health control. At the same time, the algorithm used makes it possible to group patients according to detected similarities in the constellations of health-control beliefs. Consequently, we succeeded in selecting groups that require a specific approach regarding education as compared to others. We believe that an educational intervention preceded by questionnaire studies, the results of which would divide patients into groups for which content adjusted the obtained results (e.g., in 5 versions), may be more effective than education conducted following an arbitrarily created scenario.

An interesting approach would also be to compare the obtained clustering outcomes with previous analysis of MHLC beliefs of HF patients from our published study [18]. This study used the classic statistical analysis where frequencies of health-control beliefs and self-efficacy beliefs were investigated for a group of HF patients. This classical analysis showed variability of examined psychological features, including MHLC scores, but did not end up with concrete proposals or recommendations for caring for HF patients. It is suggested that the results might be important for a partnership model of HF care for patients who acknowledge the influencing roles of others and themselves in their health management. On the contrary, the clustering results showed the beliefs about health locus of control in HF patients are not homogenous.

We believe that the clustering approach can work from the fact that such analyses are often performed in the context of marketing [51,52] where the feature-based selection of consumers sharing similar characteristics has been applied for years. In marketing, selling a product requires analyzing many parameters at the same time to send specific recommendations to the selected subgroups of potential buyers. The results indicate that it is an effective strategy to increase sales. The presented analysis was not limited to purely psychological conclusions by demonstrating the variability of selected features as well as their correlations. Our study shows that the resulting segmentation can enable practical recommendations while planning interventions for HF. We believe that on a similar basis, we can adjust the content or the way of educating the patient to their personal beliefs.

### Study Limitations

The present study had some limitations. First, the analysis was based on archived data.

Second, the study did not verify to what extent the proposed clusters were consistent with other sociodemographic data (e.g., education/acquired patients’ knowledge about HF).

Thirdly, socio-demographic, personality, or clinical variables were not included for clustering.

In sum, we are aware that the presented approach should be continued in future clustering studies, which are needed to test the effects of tailored educational interventions addressed specifically to selected clusters of HF patients.

## 5. Conclusions

In summary, the present analysis aimed to demonstrate the potential of clustering based on results reflecting structures of beliefs about health locus of control in patients with HF, which could serve as a base for precise tailoring of the patients’ education. Unlike the classic statistical analyses, our approach enables clinicians to formulate personalized recommendations for a larger sample of HF patients. This implies that advanced personalization of HF care via clustering should embrace other important psychological variables, such as depression or quality of life, which are commonly used in assessing cardiovascular patients. Thus, the implementation of clustering survey-based data from patients with chronic diseases may be a useful tool for the management of these diseases and provide practical implications in caring for patients with HF. Future studies may also be aimed at adding psychological data to clustering, based on clinical diversity in order to look at the patients from a broader perspective.

## Figures and Tables

**Figure 1 medicina-58-01542-f001:**
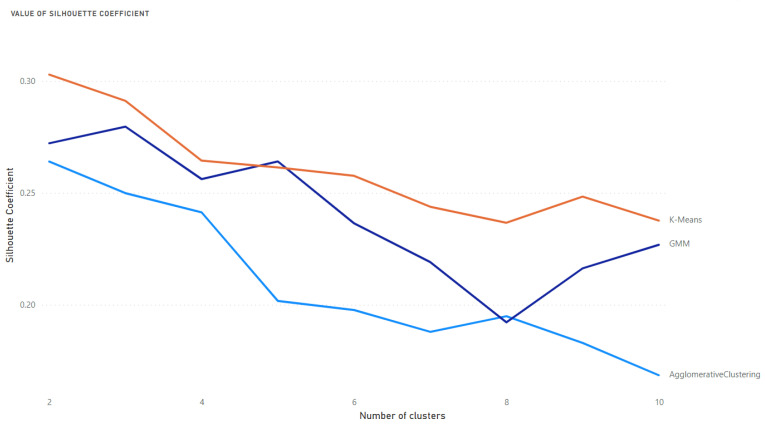
Silhouette coefficient for 3 selected algorithms and the various number of clusters.

**Figure 2 medicina-58-01542-f002:**
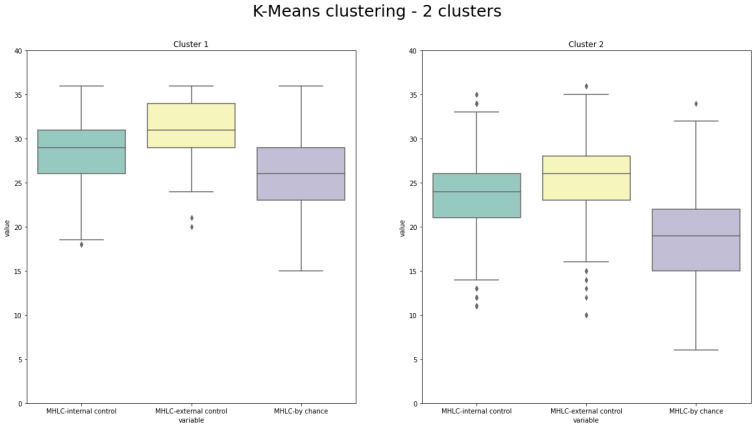
Distribution of 3 analyzed characteristics with 2 clusters generated by K-Means algorithm.

**Figure 3 medicina-58-01542-f003:**
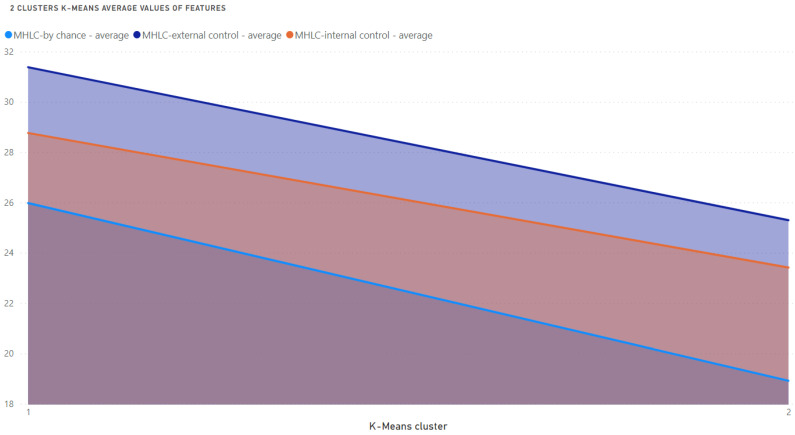
Average values of analyzed features for 2 K-Means clusters.

**Figure 4 medicina-58-01542-f004:**
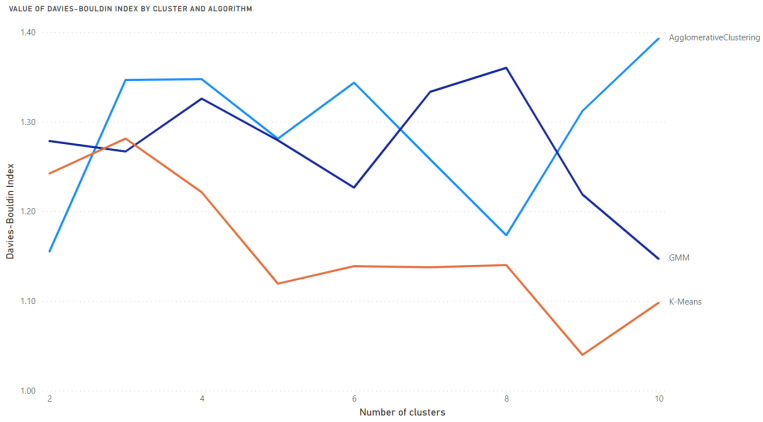
Value of the Davies-Bouldin index for 3 selected algorithms and the various number of clusters.

**Figure 5 medicina-58-01542-f005:**
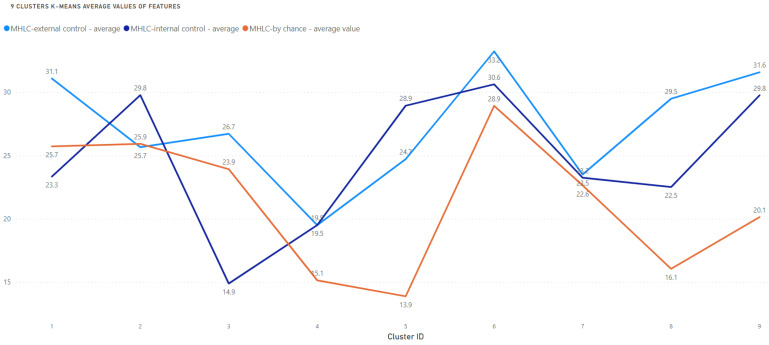
Average values of features among clusters.

**Figure 6 medicina-58-01542-f006:**
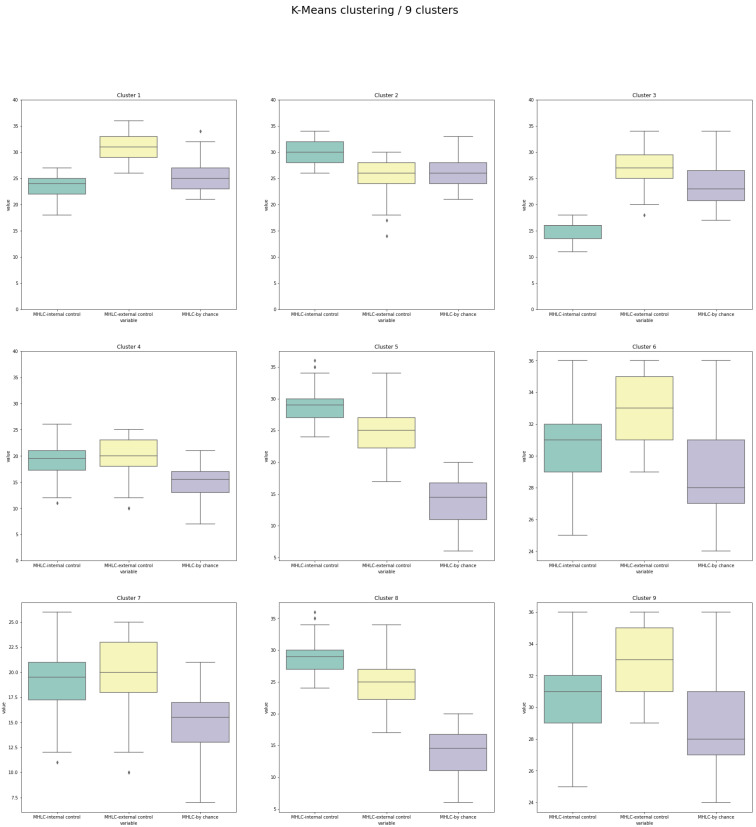
Distribution of 3 analyzed characteristics with 9 clusters generated by K-Means algorithm.

**Table 1 medicina-58-01542-t001:** Summary and comparison of 9 clusters.

Cluster	n	Internal (I) *	Others (O) *	Chance (C) *	Predicted Attitude to Education on HF Management	Observations That Should Be Encountered While Planning Educational or Motivational Interventions
1	111	3	1	2	Rather good	Risk of underestimating their role in HF management (beliefs in chance higher than internal control)
2	76	1	2	2	Problematic	Risk of ignoring professional recommendations (beliefs in chance equal to beliefs in others)
3	31	3	1	2	Very problematic	High risk of underestimating their role in HF management (the lowest internal control among all clusters)
4	42	2	1	3	Very good	Probably the most promising structure of beliefs (beliefs in others and internal control, not in chance)
5	74	1	2	3	Rather good	Risk of relying on personal, not professional recommendations (internal beliefs are the highest)
6	129	2	1	3	Rather good	The big difference between beliefs in others vs. internal control: may require very precise recommendations, risk of underestimating their role in HF management
7	118	2	1	3	Rather good	The big difference between beliefs in others vs. internal control: may require very precise recommendations, risk of underestimating their role in HF management
8	79	1	2	3	Rather good	Risk of relying on personal, not professional recommendations (internal beliefs are the highest)
9	98	2	1	3	Rather good	The big difference between beliefs in others vs. internal control: may require very precise recommendations, risk of underestimating their role in HF management

* Numbers 1–3 show which result (i.e., scores reflecting internal control (I) vs. external control by others (O) or chance (C)) was the highest (1) or the lowest (3) within each cluster.

## Data Availability

Not applicable.

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
