# Peer review of "Patterns of Locus of Control in People Suffering from Heart Failure: An Approach by Clustering Method"

_medicina, 2022, doi:10.3390/medicina58111542_

Round 1

Reviewer 1 Report (Previous Reviewer 2)

The authors have addressed the reviewer's comments in detail. The technical aspects of the cluster analysis are largely the same as those of the other reviewer. Title and abstract have been adopted as suggested. Linguistic inconsistencies have been addressed. The parameters in the formulae have been explained. The suggested text shifts have been completed. The box-plots are very illustrative.

These authors make a genuine effort to present their perspective, which comes through clearly in their replies to my comments. Since my comments are ultimately based on principles of medical ethics and the anthropology of the doctor-patient relationship, and since the present work is conceived more statistically, I decline to claim a more specific discussion of medical stance in the manuscript.

This version is acceptable for publication.

Author Response

Thank you for your valuable review.

Reviewer 2 Report (New Reviewer)

Kind authors,

your work may be interesting, but it needs further modification. from line 158 I would remove the interrogative form when you explain why you have chosen MHLC.

the method and discussions need to be reviewed. specifically, the method should be better explained. as a whole, the work in form and language must be reviewed in its entirety.

Author Response

Reviewer’s suggestion:
1. from line 158 I would remove the interrogative form when you explain why you have chosen MHLC.
Thank you very much for this suggestion. We agree that interrogative form may be considered as not suitable for scientific paper. We have changed this fragment as suggested by the reviewer:
“We decided to use MHLC, as several studies are showing that the measures of MHLC beliefs can predict compliance with medical regimes in several chronic conditions [23,51] including cardiovascular diseases [24]”.
1. the method and discussions need to be reviewed. specifically, the method should be better explained. as a whole, the work in form and language must be reviewed in its entirety
We would like to thank the reviewer for such remarks, which stimulate us to improve our work. We have re-analysed the whole manuscript in order to detect fragments which might be considered unclear or which are written in a form which is not suitable for scientific paper. We also invited some clinicians, who actually face the challenges related to HF patients’ education in their everyday clinical practice to review the paper. As a result we added some of their ideas and comments to the paper, however, according to their opinions, there was no need for improving the explanation of the applied methods. The clinical experts suggested that we should underline the fact that our paper addresses important challenges signalised in the current guidelines for HF management, according to which: insufficient self-care results from misunderstandings, misconceptions, and lack of knowledge, thus education to improve self-care should be tailored to the individual patient and based on, where available, scientific evidence or expert opinion. We believe that our model may serve as a solution leading to tailoring the education to the individual recipient (i.e. HF patient) based on professional psychological tools. Moreover, we have recently published a paper in which we demonstrated similar approach aimed at clinical data, a modern way of extracting clinically relevant differences between HF patients’ subgroups. We tried to underline that the approach in the current paper may be considered as similar or it may even broaden the exclusively-clinical approach towards looking for the features according to which patients may differ from each other. We do hope that such an extra-comments, linking our paper to the official guidelines for HF management and showing similar approach, which was applied by our group to the clinical data, will help to understand the remaining parts. In sum, as we did our best for improving the paper, we believe that it can be published in its current form. However, we are still open to receive further suggestions from the reviewer, in order to improve our message.

This manuscript is a resubmission of an earlier submission. The following is a list of the peer review reports and author responses from that submission.

Round 1

Reviewer 1 Report

The major concerns I have about this manuscript are reflected in Table 1. Cluster analysis of 758 patients using three scaled indices was performed. Based on an accepted 9-cluster solution, the average score on each of the three indices was calculated and the ordinal (rank) score for each is shown. Then, an interpretation of each cluster relative to its members’ response to educational and motivational interventions is presented. There are only six possible orderings so there is redundancy in the orderings. For example, the orderings for clusters 5 and 8 are identical and so are the interpretive comments. This is also the case for clusters 4, 6, 7, and 9. Clusters 6, 7, and 9 have identical comments but Cluster 4 is described as the most promising with respect to interventions) based on smaller difference between the internal and external influence measures. (Should not the average index values rather than the ranking be entered in the table?) The reported interpretations raise two questions. The first relates to the selection of the cluster solution. Given the overlap in interpretive comments, is the 9-cluster solution really the most meaningful? How do the cluster members differ in terms of other characteristics such as demographics to justify the distinctions? The second question relates to the interpretation of the clusters. The point of the clustering appears to be finding groups (segments) that will respond similarly to educational and motivational interventions or different types of interventions (e.g., different messaging). In this regard, the analysis presented seems highly speculative. Moreover, there is a large amount of related data in the dataset. Could some be used to provide a target classification or a proxy for one? At least, it would be of interest to see demographic and clinical profiles of the members of the clusters and how those might relate to the interpretations offered in Table 1. As noted in the discussion, education and motivating patients needs to adapt to their abilities, limitations, and predispositions. The predispositions modelled in the cluster analysis do not exist in a vacuum and their links to other characteristics are important to program design.

Minor comments:

Abstract: I have not seen the clustering methods used in the study described as “artificial intelligence-based” before. A more typical description would be “multivariate methods”.

2.6: The parameters described in lines 183 to 184 apply to the S measure, not the DB measure. Of course, those for the DB measure should be added.

It seems that the Silhouette analysis is reported and then dropped from further discussion because it provides no useful information. I think the whole analysis could be summarized in a footnote.

Reviewer 2 Report

Identifying variability in the structures of beliefs about health locus of control in patients with heart failure: a clustering approach

-        Medicina -

This manuscript is of interest for the modern conception of medicine, which does not only want to be deficit-oriented, but also resource-oriented. The term "locus of control" is both a simple and important concept to investigate the assumption of responsibility in recovery processes. The manuscript is therefore scientifically and clinically up-to-date and includes relevant (current) literature. The statistical methods are adequate but tight. Nevertheless, I do not share the rather paternalistic, normative and market-oriented human image in medicine of authors, but that is only my opinion. I reflect it by example below.

The manuscript has the quality to be published in the journal Medicina, but needs several clarifications:

1. Please try to use the English language correctly. Examples: The very first sentence of the introduction "…management which requires…" should be "…management since it requires…"; line 44 of page 1: "maintain persistence" should be "persist"; line 57 page 2   "should help to accurately tailor recommendations" should be "should help tailoring recommendations more accurately"; line 110 in page 3: “body weight” instead of “body height”; line 124 “its” instead of “their”, because you speak generically about “patient” not about patients; “balance” instead of “compromise” in line 307,  and so on.

2. Title: too long. Proposal: "Patterns of locus of control in people suffering from heart failure: an approach by clustering method".

3. Abstract: I would make the abstract, as a very important part of the paper, a bit more structured and clearer. For example, replace the first half with the following wording: "The assumption of responsibility in dealing with chronic diseases is of relevance in a resource-oriented and not only deficit-oriented medicine, especially in dealing with chronic diseases, including patients with chronic heart failure. The aim of the present study is to examine, based on the model of "locus of control", whether there are different patterns that would be relevant for a more targeted education and support of self-management in dealing with heart failure. For this purpose, a sample (N=758) from 11 Polish cardiology centres have been assessed using the standardised self-assessment scale Multidimensional Health Locus of Control (MHLC), consisting of three dimensions: i. internal localisation of health control; ii. external control by powerful others; iii. external control by chance.  Using these three criteria, nine different clusters were statistically found."

Limitations should be listed at the end, for example that no socio-demographic, personality or clinical variables were added for clustering.

4. Introduction: Throughout the paper, the authors talk about educational interventions that should be specifically tailored according to patient characteristics (extremely narrowly defined). The whole modern group theory in medicine assumes a mixture of shared characteristics (here: a chronic cardiological disease) and differences in personality, experience, resources, handling of the disease etc. (here: different locus of control); this approach is justified by the fact that on the one hand cohesion can emerge and on the other hand, because of the different styles, mutual learning, role modelling, correcting etc. can come about. This topic is very important in the discussion and should definitely be taken up.

The section between lines 77 and 81 of page 2 belongs to the method section and not to the introduction.

5. Materials and methods: what test was used to check the normal distribution of the metric variables? For formulas, please explain all letters: what is i, j,
σ, max, etc.? Please clarify: Is the Silhouette index a categorical cut-off?
How is "cooperation" defined? On which basis was "quality of cooperation" defined?

6. Results: I miss a table in which the sample is characterized on the basis of all variables. It would also be of interest (but not mandatory for this study) to have a table comparing all nine clusters for each variable (with chi-square for categorical and Scheffé post hoc test of an analysis of variance for metric variables). Otherwise, the graphs and tables listed are fine.

The following sentences belong in the Discussion and not in the Results section: "Regarding probable recommendations in the view of MHLC theory, we propose that there is only one cluster (4) of 42 patients who will be able to cooperate with professionals in the most fruitful way" (lines 272-274, page 11). Furthermore, "Such beliefs make it harder to follow advice requiring effort and engagement such as precise control over water or salt intake or regular self-monitoring" (lines 281-282, page 11). I have a lot of trouble accepting these claims, they sound very much like prejudices.

7. Discussion: many of the claims in the discussion sound paternalistic and tend to be prejudiced about what patients (should) need. I have very strong doubts that the educational groups should be composed according to the defined clusters and not heterogeneously. The results have a meaning only in the framework of a theory and this needs to be better explained. The statements about predicted co-operation with physicians (and other health professionals?) sound quite normative.

Ultimately, only (comparative) follow-up studies can determine whether one educational (group therapy) approach is better than others. Are there empirical studies about classification approaches to predicted cooperation? Is cooperation really so much important for outcome?

Following statement is much dared: "In other words, there is a need for compromise - balance? - between the group-oriented vs. individual approach. Searching for groups of patients sharing similar mental structures may be the solution". Really? We know less about real prognostic factors apart of classical medical variables. A "fruitful doctor-patient cooperation" sounds like "conformity, fitting, subordination".

Another problematic statement: "We should always expect that features responsible for being a good or bad student in childhood to be responsible for being more or less prone to education when becoming a cardiac patient. Procastination related to doing homework or forgetting about regular car check-ups may affect behaviors including regular blood pressure examinations or doctor's appointments." That is a daring utterance! Definitively, the authors share another human image in medicine than the reviewer. In the treatment of chronically ill people we deal with an anthropological based medicine and not with preference based decisions like in market studies.

The statement between line 361 and 367 (page 12) has to be moved to the "Material and methods" section.

8. Outlook: Two clusters could be of interest in assessing differences by means of logistic regressions; more clusters can be assessed only in multinomial regression models, but the interpretation is complex. Clustering is only a possible starting point for reflexion about more accurate psychosocial hypotheses and for complex statistical modelling for example by means of structural equation modelling over different measurement points. An approach to so called "complexity of human nature" (as the authors say) is only possible by means of accurate interviews followed by content analyses.